

# Uptake of gaseous formaldehyde by soil surfaces: a combination of adsorption/desorption equilibrium and chemical reactions

Guo Li[1,2], Hang Su[1], Xin Li[3], Uwe Kuhn[1], Hannah Meusel[1], Thorsten Hoffmann[4], Markus Ammann[5],

Ulrich Pöschl[1], Min Shao[2,*], Yafang Cheng[1,*]

[1]Multiphase Chemistry Department, Max Planck Institute for Chemistry, Mainz, Germany

[2]College of Environmental Sciences and Engineering, Peking University, Beijing, China

[3]Institute for Energy and Climate Research, IEK-8, Research Center Jülich, Jülich, Germany

[4]Institute of Inorganic Chemistry and Analytical Chemistry, Johannes Gutenberg University Mainz, Mainz, Germany

[5]Laboratory of Radiochemistry and Environmental Chemistry, Paul Scherrer Institute, 5232 Villigen, Switzerland

*Correspondence to*: Y. Cheng (yafang.cheng@mpic.de) or M. Shao (mshao@pku.edu.cn)





## Abstract

Gaseous formaldehyde (HCHO) is an important precursor of OH radicals and a key intermediate molecule in the oxidation of atmospheric volatile organic compounds (VOCs). Budget analyses reveal large discrepancies between modeled and observed HCHO concentrations in the atmosphere. Here, we investigate the interactions of gaseous HCHO with soil surfaces

through coated-wall flow tube experiments applying atmospherically relevant HCHO concentrations of ~10 to 40 ppbv. For the determination of uptake coefficients ($\gamma$), we provide a Matlab code to account for the diffusion correction under laminar flow conditions. Under dry conditions (relative humidity = 0%), an initial $\gamma$ of $(1.1 \pm 0.05) \times 10^{-4}$ is determined, which gradually drops to $(5.5 \pm 0.4) \times 10^{-5}$ after 8-hour experiments. Experiments under wet conditions show a smaller $\gamma$ that drops faster over time until reaching a plateau. The drop of $\gamma$ with increasing relative humidity and over time can both be explained

by the adsorption theory in which high surface coverage leads to a reduced uptake rate. The fact that $\gamma$ stabilizes at a non-zero plateau suggests the involvement of irreversible chemical reactions. Further back-flushing experiments show that two thirds of the adsorbed HCHO can be re-emitted into the gas phase while the residual is retained by the soil. This partial reversibility confirms that HCHO uptake by soil is a complex process involving both adsorption/desorption and chemical reactions which must be considered in trace gas exchange (emission or deposition) at the atmosphere-soil interface. Our

results suggest that soil and soil-derived airborne particles can either act as a source or a sink for HCHO, depending on ambient conditions and HCHO concentrations.






## 1 Introduction

Atmospheric HCHO represents one of the most abundant carbonyls in the atmosphere and is a key intermediate in atmospheric hydrocarbon oxidation. It is one of the major primary sources of $HO_x$ ($HO_x$=HO+$HO_2$) radicals (Lowe and Schmidt, 1983;Fried et al., 1997;Hak et al., 2005;Seinfeld and Pandis, 2006) and it also serves as a large source in the global budget of $H_2$ and CO (Price et al., 2007). Around 90% of global tropospheric HCHO is produced by the oxidation of methane ($CH_4$) and non-methane volatile organic compounds (NMVOCs), while direct emissions from biomass burning and fossil fuel combustion contribute to the remaining fraction (Carlier et al., 1986;Lee et al., 1998;Andreae and Merlet, 2001;Parrish et al., 2012). The known removal processes of HCHO include reaction with OH radicals, photolysis, deposition and aerosol uptake, of which the first two pathways are supposed to dominate (Zhou et al., 1996;Tie et al., 2001;Fried et al., 2003).

Budget analyses, however, reveal large discrepancies between observed HCHO concentrations and those predicted from models (Jacob, 2000;Wagner et al., 2002). Overpredictions of HCHO have been reported in a series of studies since mid-1990s (Liu et al., 1992;Heikes et al., 1996;Jacob et al., 1996;Zhou et al., 1996). For example, a recent model study at a semi-rural site in southern China by Li et al. (2014) showed 2-5 times higher HCHO than observations. On the other hand, underpredictions have also been found in a few studies (Ayers et al., 1997;Jaegle et al., 2000;Weller et al., 2000;Kormann et al., 2003). The disagreement between model calculations and observations suggests an inadequate understanding of the HCHO budget and the existence of additional source/sink terms. For example, neglecting uptake of HCHO by aerosols/clouds (Zhou et al., 1996;Tie et al., 2001;Fried et al., 2003) may result in an overestimation while insufficient consideration of NMVOCs oxidation processes may lead to underestimated production of HCHO (Wagner et al., 2002). So far the imbalance in the HCHO budget remains inconclusive.

Soil and soil-derived mineral dust could represent an important kind of surfaces regulating the budget of several trace gases and aerosols in the atmosphere (Usher et al., 2003;Kulmala and Petaja, 2011;Su et al., 2011;Oswald et al., 2013;Donaldson et al., 2014). Understanding the interactions of HCHO with soil surfaces may help to explain the discrepancies and improve our understanding of the HCHO budget. Some field flux measurements have found HCHO emission from soil surfaces (DiGangi et al., 2011) while other studies suggested a net HCHO deposition on soil surfaces (Gray et al., 2014).

In this study, we investigate the HCHO uptake on soil surfaces using a coated-wall flow tube method. The experiments are performed under conditions relevant to the atmosphere, i.e., HCHO concentrations of ~10 to 40 ppbv and relative humidities (RHs) of 0% to 70%. The uptake coefficients γ are determined by numerically solving the Cooney-Kim-Davis equation, which describes the loss of a trace species to the flow tube wall at laminar flow conditions (Murphy and Fahey, 1987). The results are discussed along with its underlying mechanisms and atmospheric implications.





## 2 Methods

### 2.1 Sample preparation

Soil samples were collected at a depth of 0-5 cm from a cultivated field site in Mainz, Germany (49°59′N, 8°13′E). The soil pH was ~7.5 (1:2 soil/water (v/v), Thermo Scientific, OrionStar A211 pH meter). The soil texture comprised 15% sand, 69% silt and 16% clay (wet sieving method) and the soil humus content was 3% (loss on ignition method) as analyzed by Envilytix GmbH (Wiesbaden, Germany). The collected samples were air-dried in the laboratory prior to grinding and sieving with a 120 mesh soil sieve. The sieved soil was autoclaved for 20 min at 394 K right before the flow tube coating procedure. The coating procedure is one of the challenges in flow tube experiments. Manually coating the tube could be time consuming and the coating thickness and homogeneity strongly depend on the operator. Here, an air-dried continuous rotating coating tool (ACRO) was developed to standardize the coating procedure and improve reproducibility and homogeneity (Fig.S.1). During the coating procedure, the tube was installed into the ACRO through two cylindrical fittings. One fitting was fixed to the inlet of the tube promoting tube rotation by means of a motor and a driving belt, while the other at the tube's outlet served only as a supporter not being fixed to the tube. The flow tube was normally placed horizontally but could also be tilted by adjusting the slope of ACRO (through a positioning screw). A drying air stream was piloted into the tube through a duct at the inlet connector.

Before coating, we prepared a soil slurry by mixing dry soil with sterilized deionized water obtained from a Milli-Q system (18.2 MΩ·cm, Millipore). The slurry was uniformly injected into an internally sandblasted glass tube which was then installed into the ACRO. The coated tube was rotated with a speed of 14 rpm and dried overnight with a flow rate of ~0.5 L min$^{-1}$ of pure $N_2$ (RH = 0%). The coating thickness/mass had been found to affect the trace gases uptake in earlier studies (Donaldson et al., 2014).  As the coating thickness increases, the uptake rate also increases due to enhanced number of surface sites available for gas uptake. Then the uptake rate moves up to a threshold level, where further increase in coating thickness doesn't affect the gas uptake. In order to exclude the influence of coating thickness, a relatively thick coating of ~500 μm was chosen for our experiments. Figure 1 shows the homogeneous soil cover thickness and structural details of the soil surface topography derived by scanning electron microscope (SEM). The good reproducibility of the experimental results as shown later also confirms the fair homogeneity and reproducibility of the coating.

### 2.2 Flow tube experiments

The uptake of HCHO onto soil surfaces was measured by employing a coated-wall flow tube system (see Fig. 2). The system consisted of four parts: (1) a HCHO generator; (2) a humidification unit; (3) a flow tube and reference tube unit; and (4) a detection unit. HCHO was generated from a solid permeation tube (formaldehyde-para, rate: 91 ng min$^{-1}$ at 60 °C, VICI Metronics Inc. U.S.A) with $N_2$ (≥ 99.999%) as the carrier gas. The RH was controlled by mixing humidified $N_2$ with generated HCHO gas.





Two glass tubes (inner surface sandblasted) with identical dimensions (length: 250 mm, i.d.: 7 mm) were used for the uptake experiments. One tube was coated with soil samples and the other remained uncoated as a reference. Both tubes were placed in a cooling jacket in which a temperature of $296 \pm 1K$ was maintained during the experiments. HCHO concentrations were

measured by an AL 4021 HCHO monitor (detection range: 100 ppt - 3 ppm, noise: 2% full scale, AERO LASER, Germany). It is an online HCHO analyzer for gaseous and liquid samples, based on the Hantzsch reaction. Gaseous HCHO is transferred into a liquid phase through a stripping coil, prior to reacting with Hantzsch reagents (ammonium acetate: for analysis, Merck, Germany; acetyl acetone: for analysis, Merck, Germany; and acetic acid (glacial): 100%, Merck, Germany). The products show a strong fluorescence at 510 nm, which is measured by a photomultiplier. The operation principle is

described in detail elsewhere (Dasgupta et al., 1988). This analyzer has been used for field measurements and has shown high stability (Preunkert et al., 2013;Preunkert et al., 2015). The flow rate used for our lab experiments was 1 L STP min$^{-1}$. The analyzer was operated with liquid calibration gas measurement mode and was calibrated using standard HCHO solutions (37 wt% in $H_2O$, Sigma-Aldrich, U.S.A).

Several flow tube experiments were carried out to examine different aspects of the uptake processes, e.g., reproducibility, RH dependence, and bi-directional exchange, etc. In each experiment, we adopted the following procedure: (1) flushing the freshly coated tube with pure $N_2$; (2) flushing the uncoated reference tube with HCHO; (3) flushing the freshly coated tube with HCHO. The first step was to detect background HCHO emission from soil. While the second step allowed for identification of potential uptake of HCHO on clean glass surface, the third step was dedicated to investigate HCHO uptake

by the soil surface, from which uptake coefficients can be derived. To check the stability of the HCHO monitor and generator, we also performed zero calibrations for the monitor and measured the generated HCHO concentrations before and after each experiment.

## 2.3 Determination of uptake coefficients

The goal of our kinetic experiments is to determine the uptake coefficients $\gamma$. The parameter $\gamma$ is defined as the fraction of

effective collisions between HCHO and the soil surface that leads to loss of HCHO due to physical or chemical processes. For investigations on trace gas uptake kinetics using flow tubes, a method, designated the CKD solution (Murphy and Fahey, 1987), is commonly applied to account for radial and axial diffusion effects. Here we develop a new method rather than using the interpolated values provided in CKD method. As our method is based on the CKD, it can be specified as CKD-based method (CKD-B). For the details of CKD-B, see Appendix A. In our experiment, the Reynolds number $R_e$ is ~200

ensuring the laminar flow conditions which require $R_e < \sim 2100$ (Murphy and Fahey, 1987). The total flow tube length is 25 cm and full development of laminar flow is achieved within ~5 cm for our setup. The uptake coefficients reported here are based on the geometric surface area of the soil sample, considering that in atmospheric models soil microstructure is not taken into account. The specific surface area of the soil sample, however, was also measured using a water vapor adsorption





method based on the Brunauer-Emmett-Teller (BET) adsorption theory (Brunauer et al., 1938) being $18.9 \pm 1.3$ m$^2$ g$^{-1}$. For calculating this BET surface area, the mass of the adsorbed water on soil sample after equilibrium with pre-defined RH levels was determined by a non-dispersive infrared (NDIR) gas analyzer (type: Li-6262, LI-COR Biosciences Inc.) operated in differential mode. This BET surface area is comparable to that in other reports on similar soil types, e.g. 12-15 m$^2$ g$^{-1}$

(Kahle et al., 2002)  and 8-19 m$^2$ g$^{-1}$ (Punrattanasin and Sariem, 2015). Accounting for the BET surface area would decrease $\gamma$ by a factor of $10^4$ in our case.

## 3 Results and discussion

### 3.1 HCHO uptake reproducibility on soil

Chamber experiments have often been used to investigate the soil emission or uptake of trace gases. A common problem of
chamber studies is that the results are not reproducible. Even for the same soil samples and experimental setup, the results may still differ from each other. This lack of reproducibility deteriorates the interpretation, extrapolation and application of experimental results.

The change of soil microbiological activities and soil micro-structure seems to be the two most probable reasons leading to
non-reproducible results. If that's the case, sterilized soil with a well-defined structure/geometry can avoid these influences and present reproducible results. To test it, we performed three uptake experiments for the same soil sample under the same RH conditions. Before each experiment, the coated flow tube was flushed with pure N$_2$ (HCHO-free) until HCHO release ceased.

As shown in Fig. 3, almost identical uptake coefficients are determined from three experiments at RH of 50% and HCHO concentration of ~35 ppbv. The reproducibility gives confidence in the kinetic parameter $\gamma$ determined from the flow tube approach.

### 3.2 HCHO uptake on soil and temporal variation

Variation of uptake coefficients with time is a key question for evaluating the atmospheric relevance of surface uptake. Fast
initial uptake can be unimportant if it quickly slows down due to consumption or occupation of reactive sites on surfaces (Kalberer et al., 1999). In this work, two 8-hour uptake experiments were performed to check the temporal variability of the HCHO uptake coefficients.

We first investigated the case under dry conditions (RH of ~0%), in which ~32 ppb of HCHO supplied by the HCHO
generator was flushed through the flow tube. As shown in Fig. 4A, the largest HCHO uptake and decrease of HCHO concentration are found in the beginning of the uptake experiment. Within the 8-hour experiment, $\gamma$ is reduced from (1.1 $\pm$



0.05) $\times 10^{-4}$ to (5.5 ± 0.4) $\times 10^{-5}$. To extend the generality of such dependence, we also conducted the same experiment but at RH of 40%. As shown in Fig. 4B, the result reveals an even faster decay of $\gamma$ over time suggesting an important role of water vapor in controlling the trace gas exchange at the atmosphere-soil interface.

### 3.3 Relative humidity (RH) effect

Water vapor has been suggested to influence the surface uptake for a variety of surface materials. On the one hand, it may compete for the reactive sites with other species and reduce their uptake (Ruiz et al., 1998;Goss et al., 2004;Donaldson et al., 2014). On the other hand, the condensed water may attract more water-soluble or hydrophilic molecules and enhance the uptake (Pei and Zhang, 2011). To better understand the role of water molecules, we examined the RH dependence of uptake coefficients. Each experiment lasted for 50 minutes with the same setup as the aforementioned 8-hour experiments.

Figure 5 shows the dependence of uptake coefficients of HCHO under different RHs. The highest value is achieved at dry conditions (RH of ~0%) and doesn't decrease much within the 50-minute experiment. Increasing RH results in a sharp decrease of $\gamma$, until reaching a RH threshold level of ~30%. Above 30% RH, $\gamma$ becomes almost independent of RH. This RH effect could be expected for a longer uptake time period (e.g., 8 hours), as the uptake coefficients always show a large

difference between dry and humid conditions during the 8-hour uptake experiments (Fig. 4A and Fig. 4B). For the detailed mechanism of the RH influence on HCHO uptake, see Sect. 3.6.

### 3.4 Concentration effect

We also investigated the dependence of the uptake coefficients on initial HCHO concentrations, under dry and humid conditions (Fig. 6). Under dry conditions, higher HCHO concentrations lead to significantly reduced uptake coefficients. The

HCHO molecules themselves exert a competition for reactive sites. This effect almost ceases under humidified conditions. As the number of water molecules is far more than that of HCHO and they both show competitive adsorption behavior on soil, it is conceivable that this concentration effect is weakened by increasing RH to 40%, further confirming the strong influence of water on the HCHO uptake.

Such a dependence on the initial concentrations of analytes under low RH conditions had already been reported by other researchers. Sassine et al. (2010) investigated the HCHO uptake kinetics on $TiO_2$ mineral coatings and found that the inverse of the uptake rate depended linearly on the inverse of the HCHO inlet concentrations. Wang et al. (2012) measured the uptake of $NO_2$ onto soil using a coated-wall flow tube and also observed that uptake coefficients were negatively dependent on $NO_2$ gas phase concentrations. They both gave explanations based on the Langmuir-Hinshelwood mechanism.





### 3.5 Reversibility of HCHO uptake

The decay of initial uptake is a common feature for trace gas uptake on different surfaces. It can be explained by different mechanisms, e.g., a process dominated by adsorption and chemical reactions; or a process governed by surface and bulk reactions. To explore the underlying mechanism, we conducted a back-flushing experiment and investigated the reversibility

of HCHO uptake.

We first performed a standard experiment by flushing HCHO through a coated flow tube with RH of 50%. Then the same tube was flushed with pure $N_2$ (HCHO free) and the amount of HCHO released elucidated potential uptake reversibility. Figure 7 shows that indeed the HCHO adsorbed by soil can be released back to the gas phase. The areas enclosed by the

uptake and emission curves (orange area $S_a$ and yellow area $S_e$) represent the amount of HCHO adsorbed and re-emitted by the soil, respectively. $S_a$ and $S_e$ could be calculated by curve fitting and integration. As the emission curve doesn't end with infinitely approaching the zero air signal, a fitted curve is used to consider this infinite decrease of HCHO signal and infer the total amount of re-emitted HCHO, reflected by $S_e$. Two functions are tested for emission curve fitting (parameters as detailed in Table 1). These fitting functions are derived on the basis of HCHO mass conservation in the flow tube, that is, the

reduced HCHO mass on the soil due to dissociation and desorption equals to that increased in the gas flow per unit time. The mass decay rate of HCHO on the soil is determined by HCHO mass $M$ and the net rate coefficient $k$, as can be expressed as $dM/dt = -k \times M^n$ ($k$ reflects the net effect of emission rate and dissociation reaction rate on the soil surface; $n$ means the apparent order of reactions between HCHO molecules (e.g. oligomerization), if $n = 1$, no reactions between HCHO molecules). Note that the HCHO mass decay mentioned here only takes into account processes which could re-emit HCHO

into the gas phase (reversible reactions) rather than those converting HCHO into other products (irreversible reactions), as the emission curve in Fig. 7 only represents the reversible fraction. By solving the above differential equation and further taking the derivative of $M(t)$, the HCHO concentration $C$ in the gas flow can be identified as a function of time $t$, as shown in Table 1 ($C = f(x)$, $t = x$). Table 1 compares two fitting functions, with the first one only accounting for zero and first order reactions and the second one further involving second order reactions. The latter provides better fitting results, implying the

existence of physical/chemical interactions between adsorbed HCHO molecules on the soil surface (e.g. hydration and oligomerization). Thus, the second fitting function is applied to infer the total amount of re-emitted HCHO. The re-emitted HCHO is $(70 \pm 15)\%$ ((average $\pm$ one standard deviation)%) of that adsorbed in the uptake experiment (Fig. 7), while the residual $(30 \pm 15)\%$ remains in the soil. This partial reversibility suggests that HCHO uptake on soil is a combination of adsorption/desorption and chemical reactions.


To further explore the reasons for this partial reversibility, we conducted an energy dispersive X-ray (EDX) analysis of the soil sample. As shown in Fig. S.2, inorganic oxides dominate the soil composition and the low fraction of carbon is consistent with the measurement of soil organic matter (Sect. 2.1). Among the inorganic oxides, silicon oxide is most



abundant followed by oxides of aluminum, calcium and iron, with their contents (wt%) being ~64%, ~13%, ~6.3% and ~5.7%, respectively. The partial reversibility can be interpreted by the different uptake ability of various components. Carlos-Cuellar et al.(2003) reported that HCHO uptake was completely reversible on $SiO_2$ but only partly (< 1~15%) reversible on α-$Al_2O_3$ and α-$Fe_2O_3$. Xu et al. (2011) investigated the heterogeneous reactions of HCHO on the surface of γ-$Al_2O_3$ particles and concluded that the adsorbed HCHO was firstly oxidized to dioxymethylene and further to formate. The fraction of silicon oxide of ~70% (silicon oxide content divided by the total amount of all inorganic oxides) in the soil investigated in here closely resembles the fraction of HCHO desorbed ((70 ± 15)%) from soil at zero air conditions.

Since $SiO_2$, $Al_2O_3$ and $Fe_2O_3$ are among the most abundant components in soils, the partial reversibility of HCHO uptake as shown in Fig. 7 can thus be expected as a general feature for various kinds of soils.

### 3.6 Uptake mechanism

Pore diffusion and surface processes could both account for the uptake of HCHO by soils. For pore diffusion within soils, the time scale depends on the thickness of the soil layer and the specific diffusion coefficients. Morrissey et al. (1999) reported typical macroscopic diffusion coefficients of VOCs ranging from $10^{-2}$ to $10^{-4}$ cm$^2$ min$^{-1}$ for clay minerals. However, the effective diffusion coefficients of VOCs in soils depend on soil properties, e.g., soil porosity, pore geometry, grain size, soil water content etc. and on VOCs characteristics, e.g., molecular size, Henry's law constant, solubility etc. (Batterman et al., 1996). Adopting the above range and utilizing Fick's second law of diffusion for our experimental case, we estimate a time scale for HCHO diffusing through the soil layer from tens of seconds to several minutes. This time scale is much less than that for our uptake experiments (8 hours), indicating that pore diffusion is not the limiting factor of uptake. In this sense, the uptake of HCHO by soil could be described by a mechanism including both adsorption/desorption and reactions of adsorbed HCHO on the soil surface.

For the adsorption process (reaction 1), gas phase HCHO, HCHO(g), can react with a reactive site S on the soil surface and become adsorbed by soil. The adsorbed HCHO(ads) can desorb into gas phase and release the reactive site. Once adsorbed, individual HCHO molecules may react with adsorbed water (hydration) and/or further combine with other HCHO to form oligomers (Toda et al., 2014). On the other hand, HCHO(ads) can also be converted to other products through chemical reactions (reaction 2).

$$HCHO(g) + S \underset{k_d}{\overset{k_a}{\rightleftharpoons}} HCHO(ads) \underset{k_p}{\overset{k_o}{\rightleftharpoons}} oligomers \tag{1}$$

$$HCHO(ads) \overset{k_r}{\rightarrow} products \tag{2}$$

$$H_2O + S \underset{k_d'}{\overset{k_a'}{\rightleftharpoons}} H_2O(ads) \tag{3}$$





H$_2$O would also undergo adsorption/desorption processes (reaction 3), occupy the reactive site and compete with HCHO. The competing effect of water molecules depends on the number of water monolayers on the soil surface. Ong and Lion (1991a) classified such effect into three regimes in their study of trichloroethylene (TCE) sorption on soil minerals. According to this mechanism, the first regime (regime I) is from dry conditions to one monolayer coverage of water on the soil surface, where direct soil-gas sorption is evident with strong competition between water and HCHO for adsorption sites on soil. The second regime (regime II) corresponds to one to five monolayers of water molecules, with likely interactions between HCHO and water including sorption of HCHO onto surface-bound water that may lead to hydration and further oligomerization of HCHO, and limited dissolution into these monolayers. The third regime (regime III) starts from approximately five layers of water molecules up to the water holding capacity of soil, where the sorption of HCHO is dominated by condensed water on soil with respective dependence on Henry's Law. In this regime, HCHO hydrates into methylene glycol or into polyoxymethylene glycols (oligomerization; Toda et al., 2014) which might further enhance the uptake of HCHO on soil.

Based on our BET experiment, one water monolayer forms at ~30% RH (Fig. 8) which is consistent with those values (20% - 30%) reported by Lammel (1999) and Goss (1993). For RH $\leq$ 30%, the HCHO sorption lies in the regime I, and water molecules show a large competing effect with HCHO as demonstrated by the strong negative dependence of $\gamma$ on the RH (Fig. 5). For RH > 30%, the water effect moves to the regime II. In this regime the adsorbed water is highly structured and modified by interactions with the soil mineral surface (Goss, 1992), and the surface area available for gas sorption is also kept constant (Goss et al., 2004). The constant surface area could explain the relatively stable uptake coefficients of HCHO observed between 30% and 70% RH as shown in Fig. 5. We haven't reached the regime III in our experiment (which would require a RH > 90%). According to Ong and Lion (1991a), the uptake in regime III follows Henry's law. So increasing RH will increase the solvent volume (or number of water molecules) on the soil surface and thus increase the adsorption capacity.

In addition to the interactions between HCHO and water molecules in different regimes mentioned above, the rates of adsorption/desorption versus the chemical reaction rates also come into play in the net exchange of HCHO.

During the uptake process, the rate expression for HCHO loss from the gas phase can be given by:

$$-\frac{d[\text{HCHO(g)}]}{dt} = k_a[\text{HCHO(g)}][\text{S}] - k_d[\text{HCHO(ads)}] \tag{4}$$

From the point of kinetic gas theory, this loss rate could also be described as:

$$-\frac{d[\text{HCHO(g)}]}{dt} = \frac{\gamma_{\text{HCHO}}\omega_{\text{HCHO}}}{4} \times \frac{A_s}{V} \times [\text{HCHO(g)}] = \frac{\gamma_{\text{HCHO}}\omega_{\text{HCHO}}}{2r}[\text{HCHO(g)}] \tag{5}$$



where $\gamma_{HCHO}$ is the uptake coefficient of HCHO, $\omega_{HCHO}$ is the thermal velocity of HCHO, $A_s$ is the geometric surface area of the soil, $V$ and $r$ are the volume and radius of the flow tube, respectively. Equating Eq. (4) and Eq. (5) yields a positive correlation between [S] and $\gamma_{HCHO}$. Increase of RH would decrease [S] due to reaction 3 and thus further reduce the value of $\gamma_{HCHO}$. This further explains the RH dependence of $\gamma$ as shown in Sect 3.3.

For the variation of adsorbed HCHO on soil surface during the uptake, the adsorption/desorption, oligomerization/dissociation and chemical reactions are all considered and the rate is given as:

$$\frac{d[HCHO(ads)]}{dt} = k_a[HCHO(g)][S] - k_d[HCHO(ads)] + (k_p - k_o)[oligomers] - k_r[HCHO(ads)] \tag{6}$$

At the start of uptake, [HCHO(ads)] is zero and only the forward reaction of adsorption is relevant. HCHO adsorbed onto the soil surface would accumulate and increase the rates of desorption, oligomerization and chemical reactions. [HCHO(ads)] will continuously grow until a steady state is established, in which case d[HCHO(ads)]/d$t$ = 0 and the net loss of HCHO in the gas phase is dominated by chemical reactions on soil. Meanwhile, the oligomerization of adsorbed HCHO may serve as a temporary buffer, storing or releasing HCHO on soil. On the other hand, the emission of HCHO from soil can also be limited
by dissociation of these oligomers and de-hydration of HCHO hydrates, this may explain the prolonged emission curve found in Fig. 7.

**4 Conclusions**

We investigated the HCHO uptake on soil surfaces by a coated-wall flow tube method. Soil exhibits strong capacity for absorbing gaseous HCHO, with initial $\gamma$ ranging from $(1.4 \pm 0.08) \times 10^{-4}$ at 0% RH to $(3.0 \pm 0.3) \times 10^{-5}$ at 70% RH based on
the geometric soil surface. Because of simultaneously acting adsorption/desorption processes, $\gamma$ shows a strong temporal dependence with an initial peak and subsequent decay until a steady state is reached. We also find a clear RH dependence of $\gamma$, especially in the low RH range (e.g. $\leq 30\%$, under ambient pressure and temperature conditions) and little RH effect at RH > 30%. The RH dependence of HCHO uptake can be explained by a transition of uptake pathways. Under low RH and water coverage (less than one monolayer) water molecules compete with gaseous HCHO for the adsorption sites on soil, while
under high RH the soil surface is fully covered with water molecules providing a near constant amount of surface adsorption sites. Besides HCHO, similar effect of RH had been reported for the soil uptake of other VOCs (Chiou and Shoup, 1985;Smith et al., 1990;Ong and Lion, 1991b, a;Pennell et al., 1992;Goss, 1993;Unger et al., 1996;Ruiz et al., 1998).

Our results also show that HCHO uptake on soil is a partial reversible process involving both adsorption/desorption and
chemical reactions (see Fig. 9). The adsorption/desorption reveals a bi-directional exchange on soil surfaces in which soil could serve either as a source or as a sink depending on ambient conditions and trace gas concentrations. Because of the



strong diurnal variability of ambient HCHO concentrations, soil water content, temperature and hence relative humidity, HCHO exchange at soil surfaces may quickly change its sign in a diurnal course as suggested for other trace gases such as HONO (Su et al., 2011;VandenBoer et al., 2015). The generality of our results on other trace gases and surfaces like aerosols still has to be elucidated.

**Acknowledgments**

This study was supported by the Max Planck Society (MPG) and National Natural Science Foundation of China (41330635). Guo Li acknowledges the financial support from the China Scholarship Council (CSC). We are grateful to Sorowka Antje for her help with the SEM and EDX analysis of the soil sample.

**Appendix A**

**Development and evaluation of the CKD-B method**

The differential equation describing the analyte (HCHO for our experimental case) concentration C as a function of axial and radial position (z, r) in a flow tube, and a new boundary condition proposed in CKD are given as:

$$u \frac{\partial C}{\partial z} = D \frac{1}{r} \frac{\partial}{\partial r} \left( r \frac{\partial C}{\partial r} \right) \tag{7}$$

$$-D \frac{\partial C}{\partial r}\bigg|_{wall} = C \frac{\bar{v}}{4} \frac{\gamma}{1 - \left( \frac{\gamma}{2} \right)} \tag{8}$$

$$-\frac{\partial C}{\partial r^*}\bigg|_{r^*=1} = N_{Shw} C \quad \text{with} \quad r^* = \frac{r}{R} \tag{9}$$

in which u is the axial flow velocity and D is the gas diffusion coefficient under experimental conditions. $\bar{v}$ is the mean molecular speed and γ is the uptake coefficient. NShw is the Sherwood number and R is the radius of the flow tube.

Equation (8) shows that Cwall (the concentration at the wall) is required to determine γ. However, the radial diffusion needs
time and may result in a concentration gradient in the radial direction. Since we can only measure the averaged concentration instead of Cwall, the determination of γ cannot be achieved unless the C profile is determined first (Behnke et al., 1997;Monge et al., 2010;Sassine et al., 2010;Kebede et al., 2013) (for details see reference Murphy et al., 1987). Therefore the factor of γ /(1- γ /2) rather than simply γ is used by Murphy et al. (1987) for a correction to the wall collision rate in the presence of a concentration gradient near the wall. Equation (9) is taken as a dimensionless form of Eq. (8).



For each given γ, a differential equation provided in CKD method (Eq. (7)) can be solved for a C profile and the functional relationship between transmittance of the analyte in the flow tube (Cout/Cin, Cout and Cin correspond to the concentrations of analyte at the outlet and the inlet of the flow tube, respectively) and γ (strictly the Sherwood number NShw as detailed in reference Murphy et al., 1987) can be established. Then by measuring the transmittance, we can determine the desired uptake

coefficients. It can be conceived that the γ - transmittance relation also depends on flow rate and the flow tube dimensions. Therefore, Murphy and Fahey (1987) provided a table of coefficients for a range of these parameters. When conditions were not available from the table, interpolation was often used in previous studies.

Here we directly solve the differential equation with numerical methods (code provided in the Supplement) rather than using

interpolated values. As this operation is based on the CKD method, it can be specified as CKD-based method (CKD-B) and this nomenclature is adopted above. More recently, another new analytical method has been derived by Knopf, Pöschl and Shiraiwa (KPS), based on a recently developed kinetic flux model framework and models discribing interactions of gas species with aerosol particles (Knopf et al., 2015) (for details see references therein). As the KPS method provides efficient and robust analyses and predictions of gas and particle uptake in flow tube experiments encountering diffusion limitation,

here, we make a comparison between CKD-B and KPS. Fig. S.3 shows that both methods agree very well for deriving the dependence of transmittance Cout/Cin on γ, under the specific flow rate and flow tube dimensions in our experimental case. To further test the accuracy of CKD-B method, the dependence of transmittance Cout/Cin on the Sherword number (NShw) is compared between CKD-B and CKD as well. Fig. S.4 shows a good agreement between results from CKD (red dots) and CKD-B (black line).

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



**List of Tables:**

**Table 1.** Parameters of fitting functions.

| Functions | Coefficients (with 95% confidence bounds) | | | | | Goodness of fit |
| --- | --- | --- | --- | --- | --- | --- |
| | $a$ | $b$ | $c$ | $d$ | $e$ | $R^2$ |
| $f(x) =$ <br> $a + b \times \exp(-c \times x)$ | $-1.18 \times 10^{-8}$ | 39.7 <br> (38.35, 41.04) | 0.616 <br> (0.607, 0.625) | | | 0.944 |
| $f(x) =$ <br> $a + b \times \exp(-c \times x) + d \times (e - d \times x)^{-2}$ | $-1.15 \times 10^{-4}$ <br> (-0.456, 0.455) | 3.13 <br> (2.35, 3.91) | 0.214 <br> (0.094, 0.333) | 0.140 <br> (0.121, 0.160) | 0.264 <br> (0.221, 0.308) | 0.998 |




**List of Figures:**

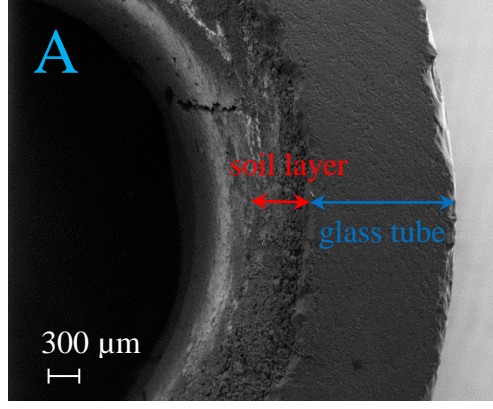

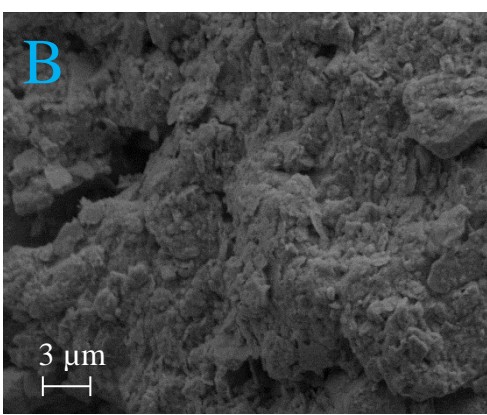

**Figure 1.** Characteristic morphology of coated soil layer observed by means of SEM (scanning electron microscope). (A) Flow tube cross-section view: the red arrow indicates the thickness of the soil layer and the blue arrow denotes the thickness of the glass tube. (B) View of the soil surface structure from above.





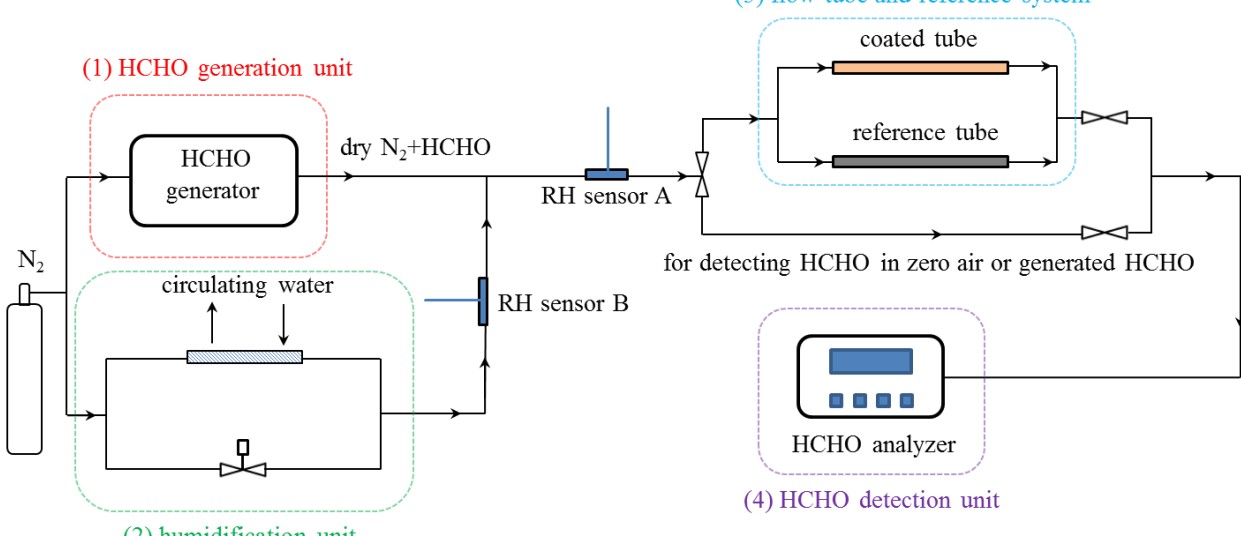

**Figure 2.** Schematic of the experimental setup. For details see text.





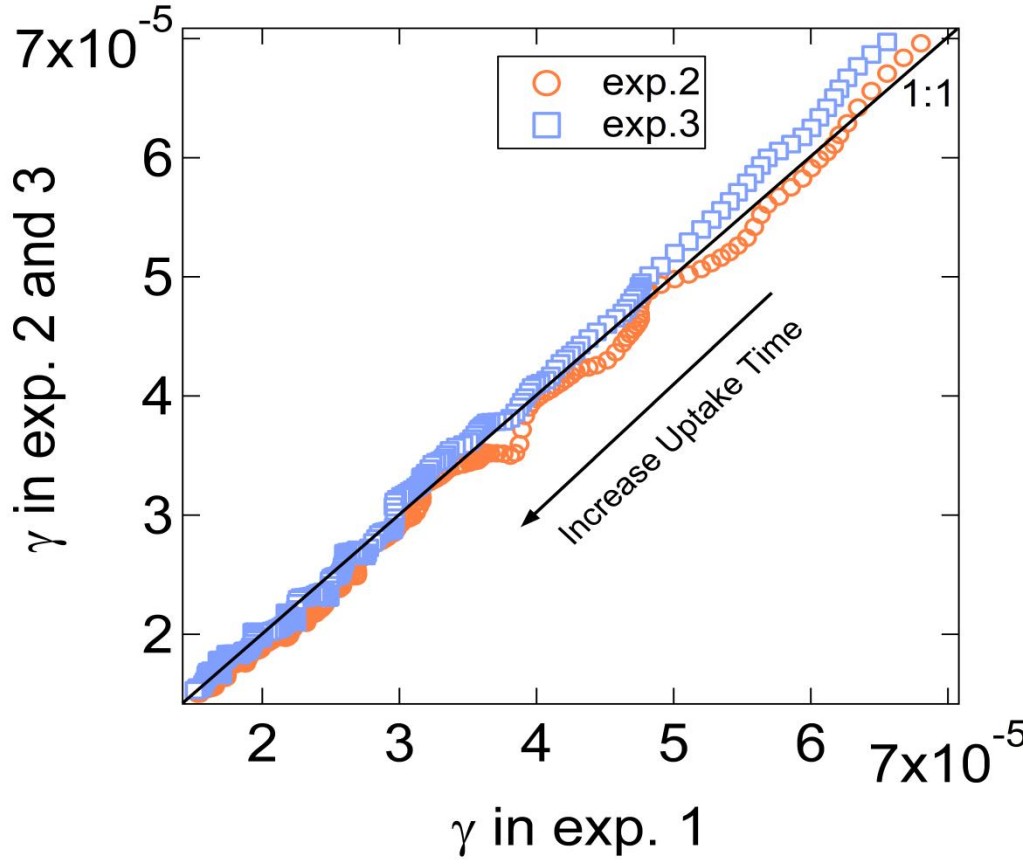

**Figure 3.** Effect of repeated uptake and emission on soil uptake coefficients within the uptake time range of 50 min at 50% RH.





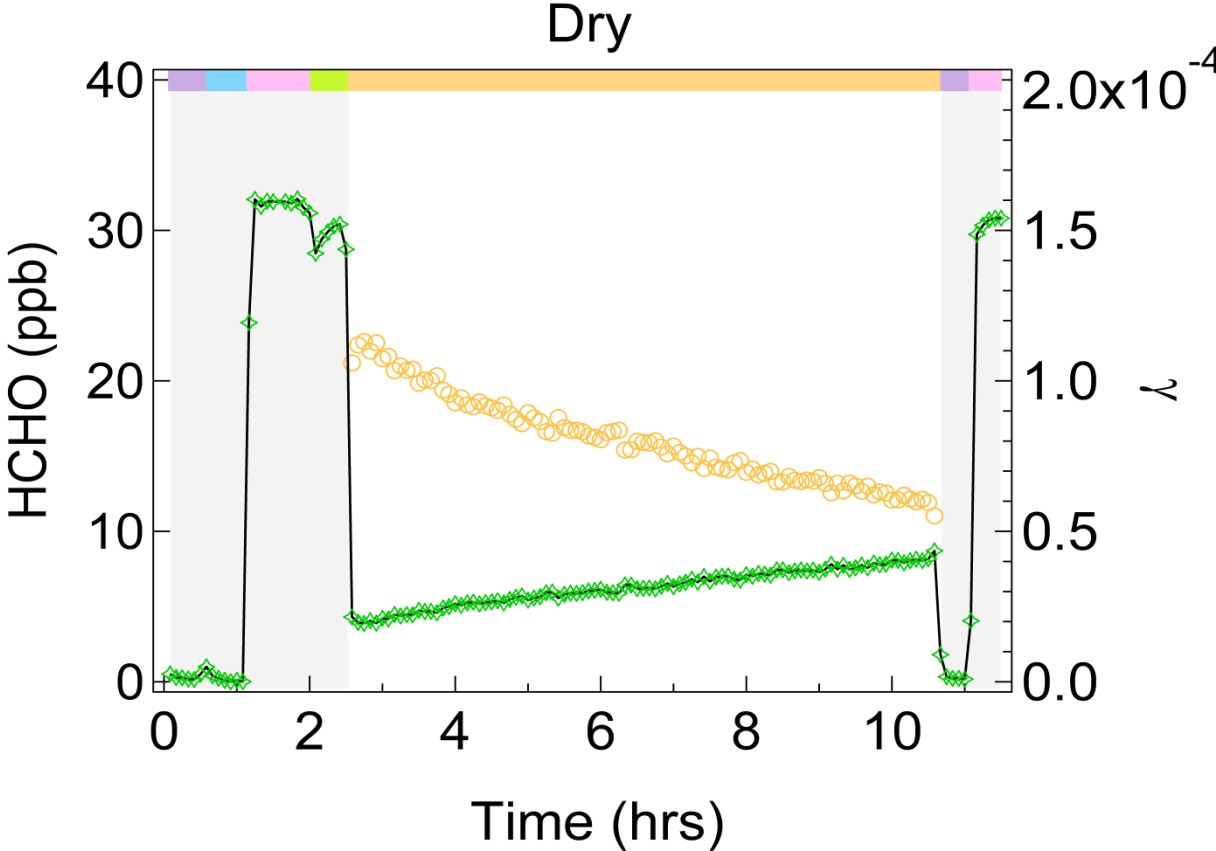

**Figure 4. (A)**





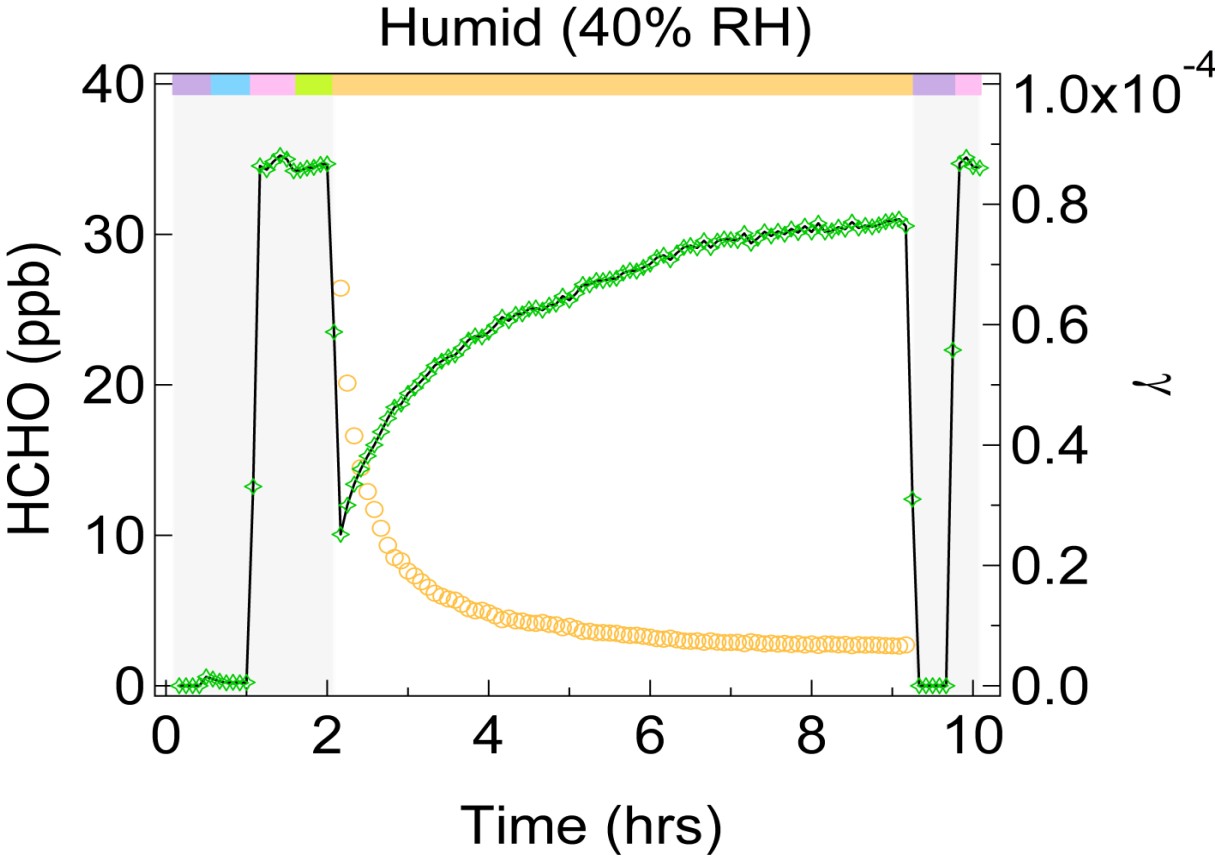

**Figure 4. (B)**

**Figure 4.** Observed HCHO (green diamonds) at the flow tube outlet and uptake coefficients (orange circles) as (A) ~32 ppb of HCHO is exposed to soil at 0% RH and (B) ~35 ppb of HCHO is exposed to soil at 40% RH. The bars with different colors indicate the time periods corresponding to different gas supply: purple bars ($N_2$), blue bar ($N_2$ after flowing through the soil-coated tube), pink bars (generated HCHO), green bar (HCHO after flowing through reference tube), orange bar (HCHO after flowing through soil-coated tube).





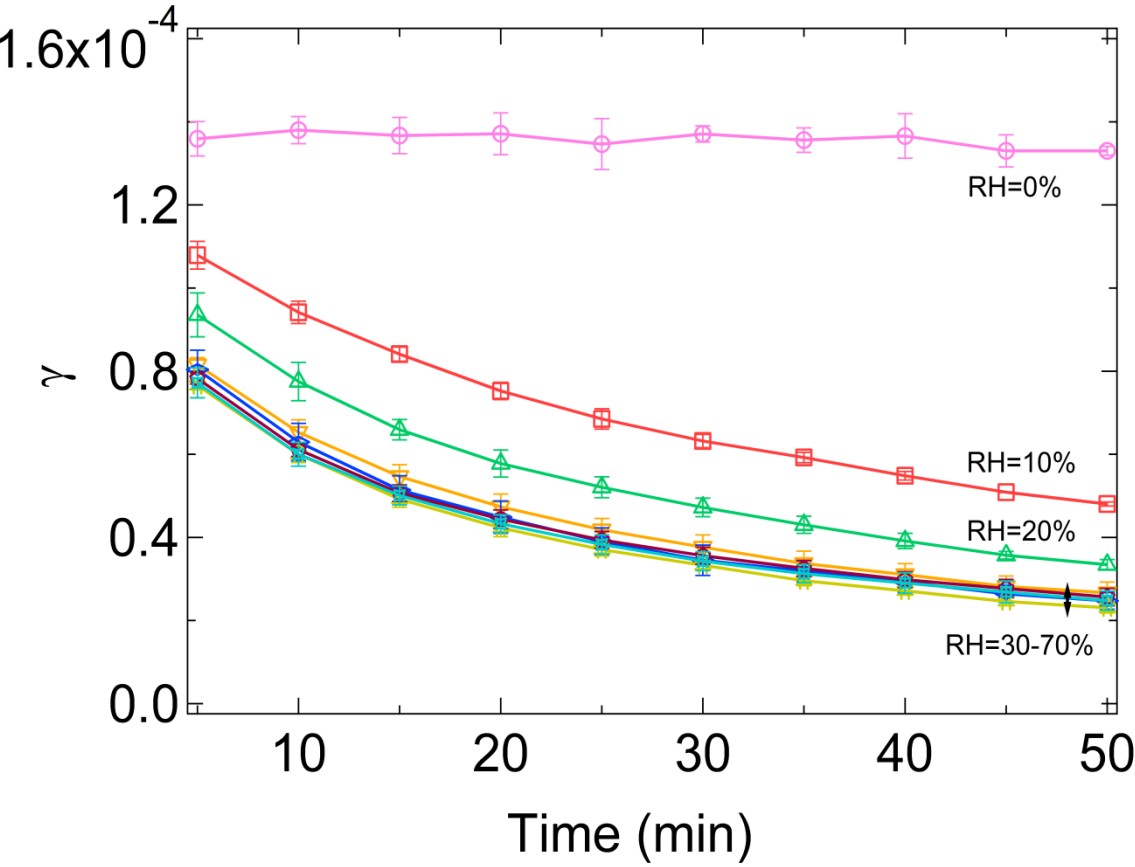

**Figure 5.** Uptake coefficients variation as a function of uptake time, under different RHs. The error bars represent the standard deviation of replicate experiments.





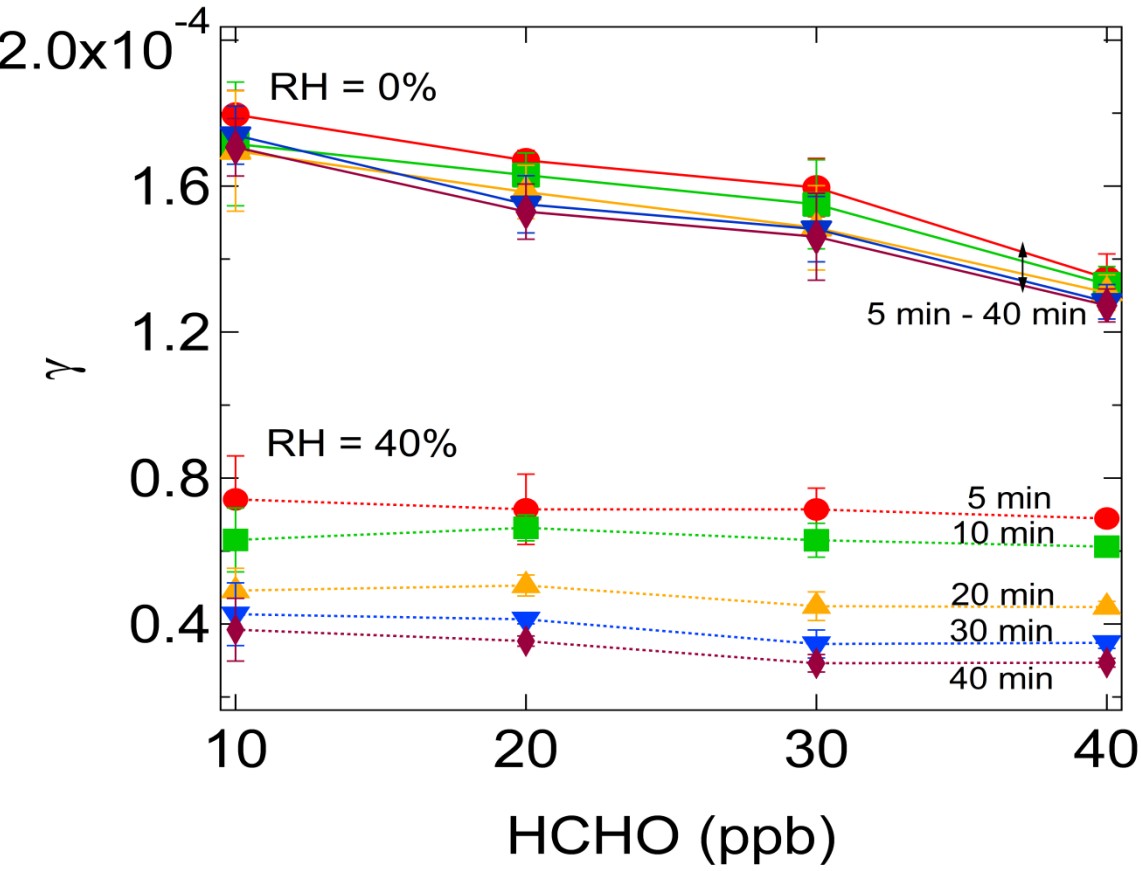

**Figure 6. (A)**




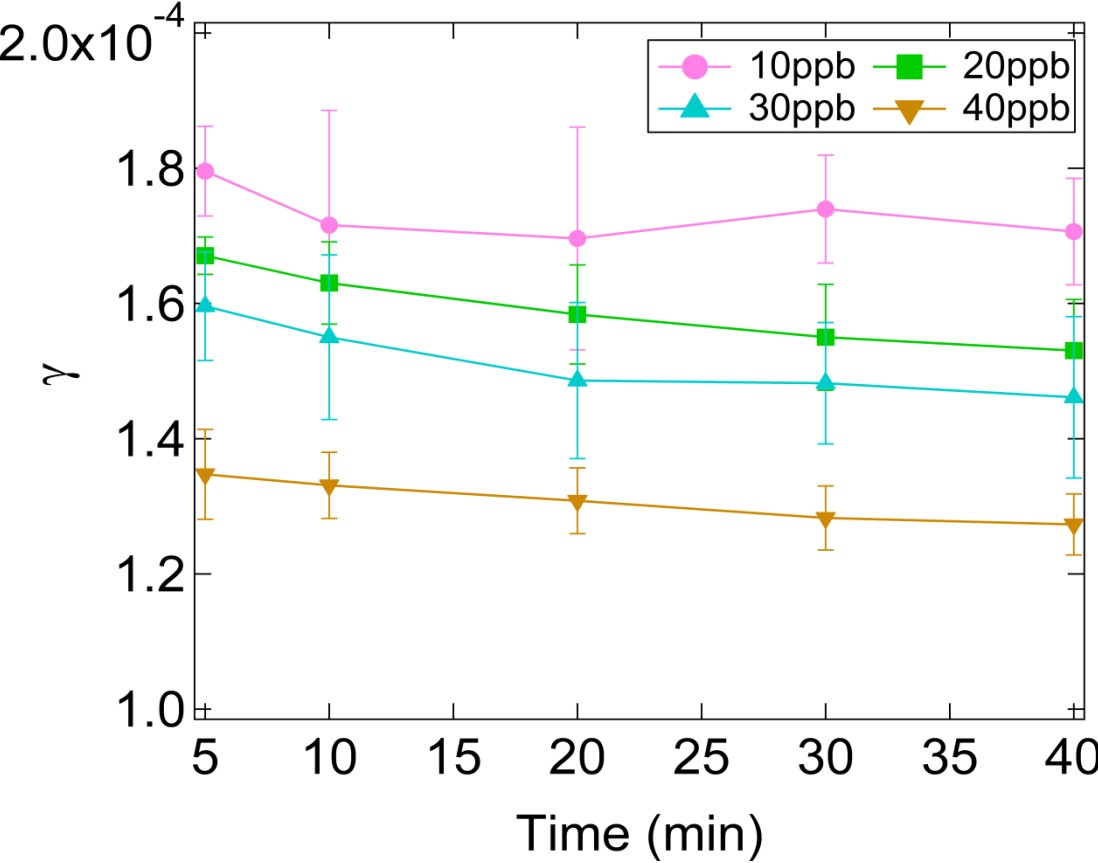

**Figure 6. (B)**

**Figure 6.** Dependence of uptake coefficients on initial HCHO concentrations at 0% and 40% RH, after uptake time periods of 5 min (red),
10 min (green), 20 min (orange), 30 min (blue) and 40 min (purple) (A); and uptake coefficients as a function of uptake time period at 0%
RH (B); both under ambient pressure and 296 K. The error bars represent the standard deviation of replicate experiments.

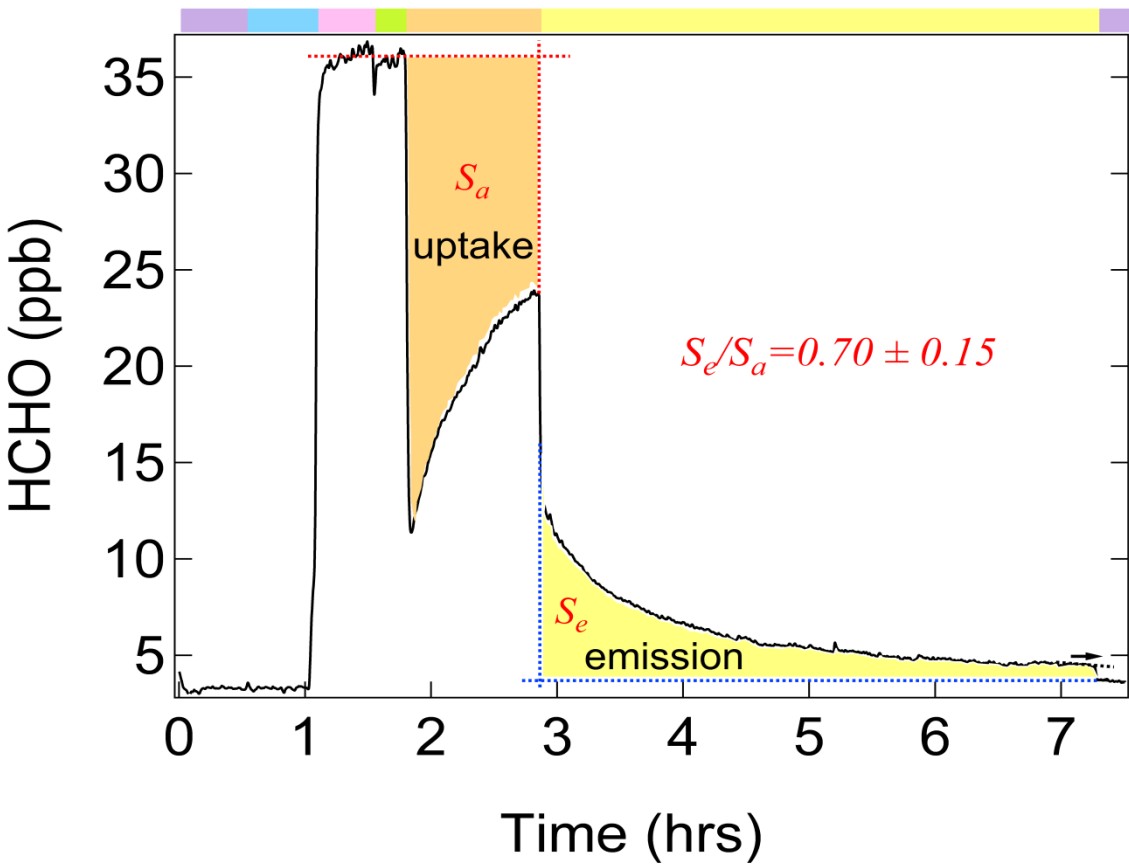

**Figure 7.** Reversible uptake of HCHO onto soil at 50% RH. The yellow bar represents the time period when flushing with $N_2$ just after the uptake experiment; other color bars as in Fig. 4.




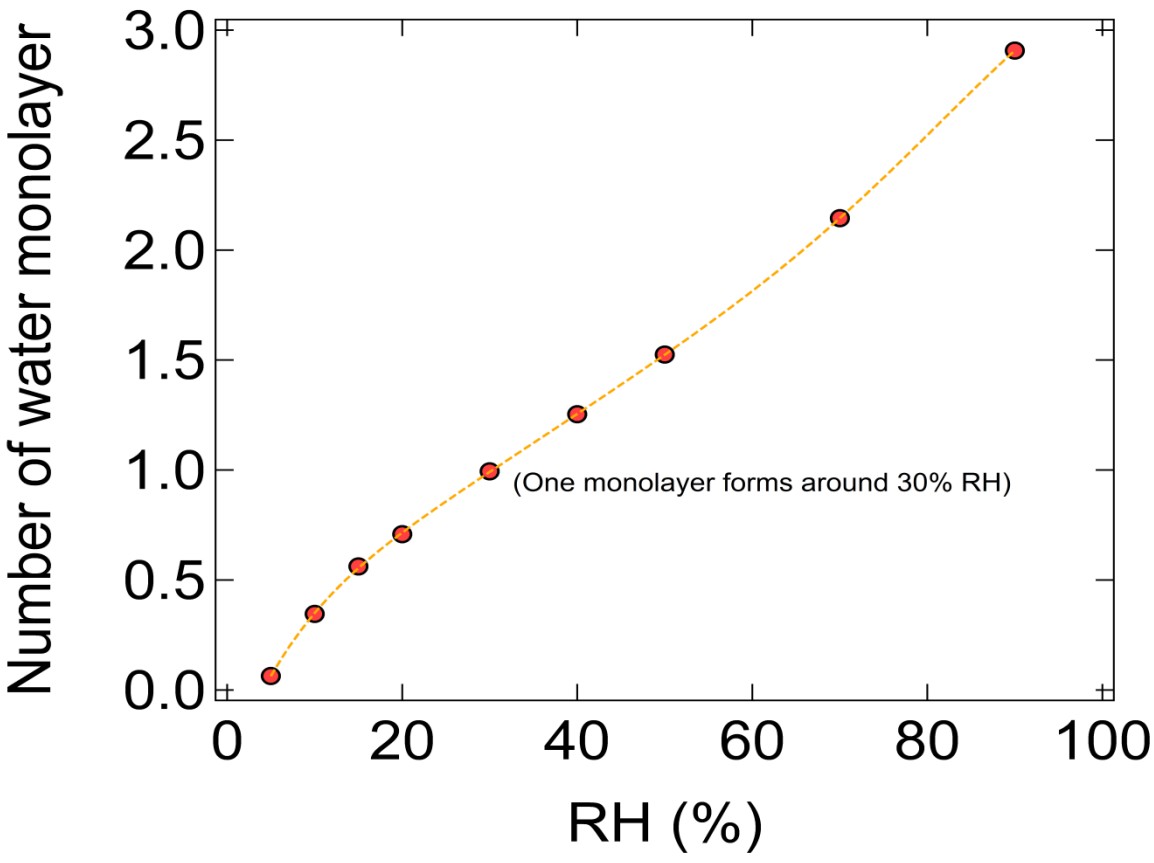

**Figure 8.** Number of soil surface water monolayers, based on gravimetric analysis of water uptake and soil surface area determined by the BET method.



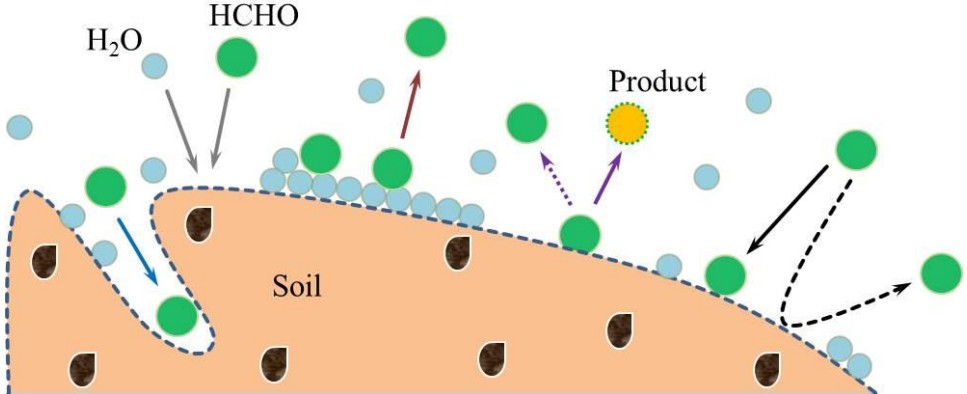

**Figure 9.** Schematic of HCHO uptake by soil surfaces.

