# Peer review of "Uptake of gaseous formaldehyde by soil surfaces: a combination of adsorption/desorption equilibrium and chemical reactions"

_Atmospheric Chemistry and Physics, 2016_

## Referee Comment (RC1) · Anonymous Referee #1 · 19 May 2016

Overall Assessment

The manuscript titled "Uptake of gaseous formaldehyde by soil surfaces: a combination of adsorption/desorption equilibrium and chemical reactions" by Li et al. describes (1) the use of coated-wall flow tube experiments to determine the uptake coefficients of formaldehyde on soil surfaces under atmospherically relevant conditions and (2) provide a chemically sound rationale for observed temporal, humidity, and reactivity trends. The mechanisms proposed make chemical sense and provides a better understanding of natural processes of formaldehyde adsorption, desorption, and reactions that were previously only partly understood. This is a very nice paper and techniques used in this study are sound and well-described. In order to improve the clarity of the article, a more

comprehensive description of the statistical methods used for data analysis would be helpful, especially how many replicates were used to determine the error bars. Also, a table of the uptake coefficients would further extend the utility of the paper as it would allow others to easily use this data for modeling. Specific comments are listed below, but the figures in general could be simplified to improve readability/interpretation.

Specific Comments

p. 7, line 29: "They both gave explanations based on the Langmuir-Hinshelwood mechanism." A brief description of the Langmuir-Hinshelwood mechanism would be helpful here for the broad readership of ACP.

p. 8, line 34: "Among the inorganic oxides, silicon oxide is most abundant followed by oxides of aluminum, calcium and iron, with their contents (wt%) being ∼64%, ∼13%, ∼6.3% and ∼5.7%, respectively." I recommend rephrasing to be more cautious with assignments. EDX data is insufficient for determining mineralogical composition. Did the authors have other data to suggest that the Si, Al, Ca and Fe are due to oxides of these elements? Some of these elements can be present as carbonates or other minerals as well.

p. 10, line 15: One monolayer at 30% RH was also reported by Donaldson et al., 2014, so there is general agreement across very different soil types.

Figure 3 would be clearer if it was a plot of uptake coefficient vs. time displaying all three experiments on the same axes.

Figure 4 might be better displayed as four separate plots instead of two compound plots as the uptake coefficient data is buried in the formaldehyde trace.

Figure 6A seems to display too much data. If the authors could make two graphs like part B, one for 0% RH and one for 40% RH, that would be easier to compare.

[Figure]

---

## Referee Comment (RC2) · Anonymous Referee #2 · 20 May 2016

This paper discusses a laboratory assessment of HCHO uptake on a soil sample using a novel soil chamber design. Recent work has begun to illustrate the potentially large atmospheric impact of soils as both a sink and source of many important trace gases. This manuscript describes laboratory experiments motivated by the need to improve our ability to parameterize the net impact of soils on global HCHO budgets. The work is very well organized and presented in a clear and concise manner. The experimental design is well thought out and for the soil type sampled a very thorough analysis of parameters such as humidity and concentration are presented. While the applicability of these measurements to real world quantification of the atmospheric impact of soil emissions would improve significantly by the addition of further experiments, I find the

paper of sufficient quality and content to warrant publication in the current form provided the comments discussed below are addressed. In particular, I believe the paper is in need of additional discussion to temper the conclusions of the paper in terms of the general applicability of the results to soil globally and even regionally, my opinions and suggestions will follow in the general comments section of this review.

General Comments:

I can imagine that the following comment will not be new to the authors but would like to see it discussed a bit further in the manuscript. The work is performed on a soil sample that has been 'sterilized' such that the effects of microbial communities can be eliminated from the experiments. In doing this only the physical and chemical properties of the soil are considered when developing a mechanistic interpretation of the results. The manner with which the authors present the results suggests that these uptake coefficients and mechanisms can be applied to a wide range, if not all soil types, and be used in as a parameterization in an atmospheric model. However, as many soil studies have shown before and the authors undoubtedly would admit by the treatment of the soil samples, addition of microbial communities have the potential to drastically alter the results of these experiments. This concept is even discussed by the authors stating that sterilization was performed to yield reproducible results. Furthermore, addition of biological litter mixed with the soils, as a soil would be expected to exist naturally, can radically alter the trace gas emission or uptake. These points make any parameterization of uptake coefficients derived from a sterile soil unrepresentative of global soils and their potential impact on atmospherically relevant gases. The authors need to put their results in context in a much more conservative manner explicitly identifying the limitation of utilizing this overly simplistic soil laboratory model. Furthermore, in using only a single soil claims about the applicability of these observations to other soil systems are a stretch and not backed by the laboratory data. This paper is a fine example of how these types of experiments should be performed and certainty presents a novel technique and a unique and useful data set, however I feel the conclusions attempt

to extrapolate beyond the capacity of the experiments presented. As presented, the results here have very little relevance to the atmospheric impact of soils on HCHO.

Specific Comments:

Calibrations for this instrument were done using liquid standards, were their any corrections applied to the data to account for removal efficiencies of the scrubber? Would there be differences in the calibration if it were done with a gas phase standard? How is the measurement potentially affected by the changing humidity, e.g. is there a humidity dependence in the HCHO measurement? I understand that these experiments were likely performed when flowing clean N2 over the coated reactor, but can the authors include a statement on how the instrument background changed with relative humidity? If there are background humidity effects these could potentially be amplified with the addition of HCHO.

The statement "the partial reversibility of HCHO uptake as shown in Fig. 7 can thus be expected as a general feature for various kinds of soils." is a conclusion based on the authors support of the idea that most soils share a similar metal oxide composition. However, I am not entirely sure the claim of various soils showing similar reversibility can be made considering the mechanism the author discusses within this paper do not deal with the chemical composition of the soil, but seem more controlled by surface area effects which could be expected to vary largely between soil types. Surely other factors such as soil pH and surface morphology of a given soil would have a more dominant control on the reversibility of uptake than the metal oxide composition.

Page 7, line 19: "Under dry conditions, higher HCHO concentrations", in this case what is the higher HCHO concentrations used? Page 8, line 15: It seems this sentence is in need of an edit as it reads awkwardly: "...dissociation and desorption equals to that increased in the gas flow per unit time."

Page 8, line 16: delete the word 'as' after "coefficient k,".

Page 9, line 6: delete the word 'in' after "the soil investigated"

Page 10, line 20: delete the word 'the' before "regime III".

Page 11, line 26: edit to read "a similar effect" Page 11, line 29: edit "partial" to 'partially'

―――――――――――――――――――――

---

## Author Comment (AC1) · 14 Jul 2016

**Response to Anonymous Referee #1**

We thank the reviewer for the constructive suggestions/comments. Below we provide a point-by-point response to individual comments (reviewer comments and suggestions are in italics, responses and revisions are in plain font; revised sections in the manuscript text in response to the comments are marked with red color; page numbers refer to the ACPD version; figures and tables used in the responses are labeled as Fig. R1, Table R1, …).

**Comments and suggestions:**

*In order to improve the clarity of the article, a more comprehensive description of the statistical methods used for data analysis would be helpful, especially how many replicates were used to determine the error bars.*

**Responses and Revisions:**

Good suggestion. We have revised the captions of Figure 5 and Figure 6 (as Fig. R5 and Fig. R6, respectively), and indicated the number of replicate experiments used for error bars:

[Figure]

**Figure R5.** Uptake coefficients variation as a function of uptake time, under different RHs. The error bars represent the standard deviation of three replicate experiments.

[Figure]

**Figure R6 (A).**

[Figure]

**Figure R6 (B).**

**Figure R6.** Dependence of uptake coefficients on initial HCHO concentrations at 0% and 40% RH, after uptake time periods of 5 min (red), 10 min (green), 20 min (orange), 30 min (blue) and 40 min (purple) (A); and uptake coefficients as a function of uptake time period at 0% RH (B);

both under ambient pressure and 296 K. The error bars represent the standard deviation of three replicate experiments.

**Comments and suggestions:**

*Also, a table of the uptake coefficients would further extend the utility of the paper as it would allow others to easily use this data for modeling.*

**Responses and Revisions:**

Good suggestion. We have summarized the uptake coefficients in a table in the supplemental information (Table S.1.) to allow others to easily use the data for modeling, which is now referred to on page 6, line 22:

"…In order to facilitate modelling, the calculated uptake coefficients as a function of initial HCHO concentration, relative humidity and uptake time period are summarized in Table S.1."

| Initial HCHO con. $C_{in}$ (ppb) | RH (%) | $\gamma^a \times 10^{-5}$ | $\gamma^b \times 10^{-5}$ | $\gamma^c \times 10^{-5}$ | $\gamma^d \times 10^{-5}$ | $\gamma^e \times 10^{-5}$ | $\gamma^f \times 10^{-5}$ |
|---|---|---|---|---|---|---|---|
| 10 | 0 | 17.9 ± 0.7 | 17.2 ± 1.7 | 17.0 ± 1.6 | 17.4 ± 0.8 | 17.1 ± 0.8 | 16.7 ± 1.3 |
| 10 | 40 | 7.4 ± 1.2 | 6.3 ± 0.9 | 4.9 ± 0.6 | 4.3 ± 0.8 | 3.8 ± 0.9 | 3.5 ± 0.8 |
| 20 | 0 | 16.7 ± 0.3 | 16.3 ± 0.6 | 15.8 ± 0.7 | 15.5 ± 0.8 | 15.3 ± 0.8 | 15.4 ± 0.4 |
| 20 | 40 | 7.2 ± 1.0 | 6.6 ± 0.4 | 5.1 ± 0.3 | 4.1 ± 0.1 | 3.5 ± 0.1 | 3.2 ± 0.1 |
| 30 | 0 | 16.0 ± 0.8 | 15.5 ± 1.2 | 14.8 ± 1.2 | 14.8 ± 0.9 | 14.6 ± 1.2 | 14.2 ± 0.9 |
| 30 | 10 | 11.6 ± 0.3 | 9.4 ± 0.3 | 7.5 ± 0.2 | 6.3 ± 0.2 | 5.5 ± 0.1 | 4.8 ± 0.1 |
| 30 | 20 | 10.4 ± 0.5 | 7.8 ± 0.5 | 5.8 ± 0.3 | 4.7 ± 0.2 | 3.9 ± 0.2 | 3.3 ± 0.1 |
| 30 | 30 | 9.4 ± 0.3 | 6.5 ± 0.3 | 4.7 ± 0.3 | 3.8 ± 0.3 | 3.1 ± 0.3 | 2.7 ± 0.2 |
| 30 | 40 | 7.1 ± 0.6 | 6.3 ± 0.5 | 4.5 ± 0.4 | 3.4 ± 0.4 | 2.9 ± 0.2 | 2.5 ± 0.2 |
| 30 | 50 | 7.6 ± 0.3 | 6.0 ± 0.2 | 4.2 ± 0.2 | 3.3 ± 0.1 | 2.7 ± 0.1 | 2.3 ± 0.1 |
| 30 | 60 | 7.8 ± 0.1 | 6.1 ± 0.2 | 4.4 ± 0.2 | 3.6 ± 0.2 | 3.0 ± 0.2 | 2.6 ± 0.2 |
| 30 | 70 | 7.7 ± 0.4 | 6.0 ± 0.3 | 4.3 ± 0.2 | 3.4 ± 0.2 | 2.9 ± 0.2 | 2.5 ± 0.2 |
| 40 | 0 | 13.5 ± 0.7 | 13.3 ± 0.5 | 13.1 ± 0.5 | 12.8 ± 0.5 | 12.7 ± 0.4 | 12.7 ± 0.3 |
| 40 | 40 | 6.9 ± 0.1 | 6.1 ± 0.2 | 4.5 ± 0.2 | 3.5 ± 0.2 | 2.9 ± 0.1 | 2.5 ± 0.1 |

[a]Uptake coefficients at uptake time period of 5 min. [b]10 min. [c]20min. [d]30min. [e]40min. [f]50min.
The error bars represent one standard deviation of three replicates.

**Table R.S.1.** Calculated HCHO uptake coefficients as a function of initial HCHO concentration, relative humidity and uptake time period.

**Comments and suggestions:**

*p. 7, line 29: "They both gave explanations based on the Langmuir-Hinshelwood mechanism." A brief description of the Langmuir-Hinshelwood mechanism would be helpful here for the broad readership of ACP.*

**Responses and Revisions:**

Good suggestion. It is necessary to give a description of the Langmuir-Hinshelwood mechanism to help the broad readership understand it more easily. Therefore, we have added a brief description concerning this mechanism on page 8, line 23:

"…They both gave explanations based on the Langmuir-Hinshelwood mechanism, in which gas molecules compete for the adsorption sites and the adsorbed molecules undergo following reactions. At higher HCHO concentrations (20 − 40 ppb) or/and higher water vapour partial pressure (RH = 40%), the soil surface becomes more easily covered by adsorbed HCHO or/and $H_2O$ molecules that had not yet reacted, resulting in lower probability of successful collisions between HCHO and the soil surface and thus a lowering of the uptake coefficient."

**Comments and suggestions:**

*p. 8, line 34: "Among the inorganic oxides, silicon oxide is most abundant followed by oxides of aluminum, calcium and iron, with their contents (wt%) being ~64%, ~13%, ~6.3% and ~5.7%, respectively." I recommend rephrasing to be more cautious with assignments. EDX data is insufficient for determining mineralogical composition. Did the authors have other data to suggest that the Si, Al, Ca and Fe are due to oxides of these elements? Some of these elements can be present as carbonates or other minerals as well.*

**Responses and Revisions:**

The reviewer was right. The elements of Si, Al, Ca and Fe can be present as several different forms (i.e., as oxides, carbonates or other minerals). Unfortunately, we do not have any further information on the determination of mineralogical composition. So we have rephrased this section to be more scientifically rigorous (page 9, line 27):

"…As shown in Fig. S.2, inorganic elements (mainly exist as minerals) dominate the soil composition and the low fraction of carbon is consistent with the measurement of soil organic matter (Sect. 2.1). Among the inorganic elements, silicon is most abundant followed by aluminium, calcium and iron, with their contents (wt%) being ~64%, ~13%, ~6.3% and ~5.7%,

respectively. The partial reversibility can be interpreted by the different uptake ability of various components. Carlos-Cuellar et al.(2003) reported that HCHO uptake was completely reversible on $SiO_2$ but only partly (< 1~15%) reversible on $\alpha\text{-}Al_2O_3$ and $\alpha\text{-}Fe_2O_3$. Xu et al. (2011) investigated the heterogeneous reactions of HCHO on the surface of $\gamma\text{-}Al_2O_3$ particles and concluded that the adsorbed HCHO was firstly oxidized to dioxymethylene and further to formate. The fraction of silicon of ~70% (silicon content divided by the total amount of all inorganic elements) in the soil investigated here closely resembles the fraction of HCHO desorbed ((70 ± 15)%) from soil at zero air conditions.

Since mineral particles occupy the major volume (approximately 45% - 49%) of soils (DeGomez et al., 2015) and silicon minerals (e.g., silicon oxides) are fairly common in mineral particles, the partial reversibility of HCHO uptake as shown in Fig. 7 may be expected as a general feature for other similar types of soils.

**Comments and suggestions:**

*p. 10, line 15: One monolayer at 30% RH was also reported by Donaldson et al., 2014, so there is general agreement across very different soil types.*

**Responses and Revisions:**

Good suggestion. We have added Donaldson's paper as a reference and found the fact that one monolayer formed around 20-30% RH could apply across very different soil types (page 11, line 12):

"…Based on our BET experiment, one water monolayer forms at ~30% RH (Fig. 8) which is consistent with those values (20%-30%) reported by Donaldson et al. (2014), Lammel (1999) and Goss (1993) retrieved across different soil types."

**Comments and suggestions:**

*Figure 3 would be clearer if it was a plot of uptake coefficient vs. time displaying all three experiments on the same axes.*

**Responses and Revisions:**

Good suggestion. We have re-plotted Figure 3 in a form of uptake coefficient vs. time and displayed all three experiments on the same axes. Also, the uptake time of each experiment has been stated in page 7, line 14:

"As shown in Fig. 3, almost identical uptake coefficients are determined from three experiments (each lasted for 50 minutes) at RH of 50% and HCHO concentration of ~35 ppbv."

[Figure]

**Figure R3.** Effect of repeated uptake and emission on soil uptake coefficients within the uptake time range of 50 min at 50% RH.

**Comments and suggestions:**

*Figure 4 might be better displayed as four separate plots instead of two compound plots as the uptake coefficient data is buried in the formaldehyde trace.*

**Responses and Revisions:**

We thank the reviewer's good suggestion. The uptake coefficient data shown in Figure 4 is tended to indicate its variation trend as a function of uptake time (see the results and discussion part). As we have further provided an uptake coefficient table (according to the reviewer's suggestion), the exact uptake coefficient values are now available for readers who are interested in using the data. In this sense, we would like to keep Figure 4 in its original version.

**Comments and suggestions:**

*Figure 6A seems to display too much data. If the authors could make two graphs like part B, one for 0% RH and one for 40% RH, that would be easier to compare.*

**Responses and Revisions:**

We thank the reviewer's good suggestion. Figure 6A is aimed to compare the dependence of uptake coefficient on initial HCHO concentration between dry and humid (RH=40%) conditions. It is more comparable, in our opinion, if both conditions are shown in one plot, as we can also see the effect of water vapor on decreasing the uptake coefficient. For the case of RH=0% in Figure 6A, the symbols are overlapped and the respective information can be retrieved in Figure 6B. So we would tend to keep Figure 6A in its original version.

**References:**

Carlos-Cuellar, S., Li, P., Christensen, A. P., Krueger, B. J., Burrichter, C., and Grassian, V. H.: Heterogeneous uptake kinetics of volatile organic compounds on oxide surfaces using a Knudsen cell reactor: Adsorption of acetic acid, formaldehyde, and methanol on alpha-Fe2O3, alpha-Al2O3, and SiO2, J Phys Chem A, 107, 4250-4261, 2003.

Basic Soil Components: http://articles.extension.org/pages/54401/basic-soil-components, 2015.

Donaldson, M. A., Berke, A. E., and Raff, J. D.: Uptake of Gas Phase Nitrous Acid onto Boundary Layer Soil Surfaces, Environ Sci Technol, 48, 375-383, 2014.

Goss, K. U.: Effects of Temperature and Relative-Humidity on the Sorption of Organic Vapors on Clay-Minerals, Environ Sci Technol, 27, 2127-2132, 1993.

Lammel, G.: Formation of Nitrous Acid: Parameterisation and Comparison with Observations.286, 1999.

Xu, B. Y., Shang, J., Zhu, T., and Tang, X. Y.: Heterogeneous reaction of formaldehyde on the surface of gamma-Al2O3 particles, Atmos Environ, 45, 3569-3575, 2011.

---

## Author Comment (AC2) · 14 Jul 2016

**Response to Anonymous Referee #2**

We thank the reviewer for the constructive suggestions/comments. Below we provide a point-by-point response to individual comments (reviewer comments and suggestions are in italics, responses and revisions are in plain font; revised sections in the manuscript text in response to the comments are marked with red color; page numbers refer to the ACPD version; figures and tables used in the responses are labeled as Fig. R1, Table R1, …).

**Comments and suggestions:**

*I can imagine that the following comment will not be new to the authors but would like to see it discussed a bit further in the manuscript. The work is performed on a soil sample that has been 'sterilized' such that the effects of microbial communities can be eliminated from the experiments. In doing this only the physical and chemical properties of the soil are considered when developing a mechanistic interpretation of the results. The manner with which the authors present the results suggests that these uptake coefficients and mechanisms can be applied to a wide range, if not all soil types, and be used in as a parameterization in an atmospheric model. However, as many soil studies have shown before and the authors undoubtedly would admit by the treatment of the soil samples, addition of microbial communities have the potential to drastically alter the results of these experiments. This concept is even discussed by the authors stating that sterilization was performed to yield reproducible results. Furthermore, addition of biological litter mixed with the soils, as a soil would be expected to exist naturally, can radically alter the trace gas emission or uptake. These points make any parameterization of uptake coefficients derived from a sterile soil unrepresentative of global soils and their potential impact on atmospherically relevant gases. The authors need to put their results in context in a much more conservative manner explicitly identifying the limitation of utilizing this overly simplistic soil laboratory model. Furthermore, in using only a single soil claims about the applicability of these observations to other soil systems are a stretch and not backed by the laboratory data. This paper is a fine example of how these types of experiments should be performed and certainty presents a novel technique and a unique and useful data set, however I feel the conclusions*

*attempt to extrapolate beyond the capacity of the experiments presented. As presented, the results here have very little relevance to the atmospheric impact of soils on HCHO.*

**Responses and Revisions:**

We thank the reviewer's very detailed comments and considerations relevant to the experimental design and the interpretation and extrapolation of our experiment results. Based on the reviewer's comments that microbial communities may largely change the soil uptake properties for HCHO, we further conducted two uptake experiments using natural soil (unsterilized) and sterilized soil, respectively, under HCHO concentration of ~ 30 ppb, room temperature and RH of 70% conditions. Both the type of soil and the experimental procedure were the same as described in the manuscript. The uptake experiments lasted for 12 hours and the uptake coefficients were calculated at each hour, as shown in Figure R.1. From Fig. R.1, the uptake coefficient between unsterilized soil and sterilized soil doesn't show much difference during the whole uptake time period, suggesting for our case the microbial effect on HCHO uptake is small. This microbial effect, however, may also depend on the soil type and the experimental conditions applied.

[Figure]

**Figure R.1.** Uptake coefficient variation as a function of time, for both unsterilized and sterilized soil samples.

In agreement with the reviewer's suggestions, we now better emphasize on page 4, line 8:

"Different soil types are inhabited by different microbial communities being very sensitive to soil properties (e.g., soil water content, pH, temperature, etc.). The applied wide range of changes in relative humidity within our experiments affects the soil water content, with respective impact on microbial activity. For a natural soil probe, the apportionment between the soil microbiological trace gas exchange and soil physicochemical effects would become vague, specifically in view of time-dependent patterns of microbial activity after changes in soil water content. The soil sterilization treatment eliminates the effects of microbiological activity on trace gas uptake/emission mechanism. The soil samples used in our study may serve as a soil proxy to study the physicochemical side of trace gas exchange, constituting a key element in regulating trace gas exchange at the atmosphere-soil interface (Donaldson et al., 2014;VandenBoer et al., 2015). However, our results on sterilized soil samples cannot necessarily be considered representative for other and/or natural types of soils."

**Comments and suggestions:**

*Calibrations for this instrument were done using liquid standards, were there any corrections applied to the data to account for removal efficiencies of the scrubber? Would there be differences in the calibration if it were done with a gas phase standard? How is the measurement potentially affected by the changing humidity, e.g. is there a humidity dependence in the HCHO measurement? I understand that these experiments were likely performed when flowing clean $N_2$ over the coated reactor, but can the authors include a statement on how the instrument background changed with relative humidity? If there are background humidity effects these could potentially be amplified with the addition of HCHO.*

**Responses and Revisions:**

We thank the reviewer's comments.

Indeed, using liquid standards for the instrument calibration, no corrections were applied to account for the scrubber removal efficiency, which was shown to be ≥98% for this kind of commercially available instrument (Krinke, 1999).

Biases to the instrument's response can be expected if either not all the HCHO in gas phase can

be stripped by the stripper (i.e., stripping efficiency < 100%), or if the water vapor in the gas flow has  dilution/concentration  effects on the stripping solution. At the outlet of the stripping coil, the sample air is saturated to 100% RH at 10°C. Depending on the inlet air RH (at room temperature) the stripping solution will gain (condensation) or lose (evaporation) water according to the gas/liquid phase equilibrium (Junkermann and Burger, 2006). The bias in calculated HCHO concentrations is within ± 3%.  To check the RH effect on measured HCHO concentration under varying RH conditions, we occasionally compared the observed concentrations with data derived from the certified permeation rate of the permeation tube and the measured gas flow rate (gas phase calibration method). The observed total bias, assumed to characterize the instrument's accuracy was within ± 5%. Moreover, the instrument's zero air background was observed to increase gradually when increasing RH from 0% to 50%, and decreased when further increasing the RH beyond 70%. This could be due to the RH effect on the measurement principle, inlet surface effects, or aging of peristaltic tubes, stripping solution and Hantzsch solution (Kaiser et al., 2014). To account for the influence of changed background on measured HCHO concentration, the background was checked at the beginning and end of each experiment and the measured HCHO concentrations were corrected based on observed background values.

On the other hand, well-known ozone interferences (Rodier and Birks, 1994;Kormann et al., 2003) were not of any concern for our studies, as the applied carrier gas was free of ozone.

In any case, we have added a statement concerning the relative humidity effect on instrument background and measured HCHO signals, on page 5, line 30:

"…As a wide range of RH conditions (0% - 70%) was applied in the uptake experiments, the potential effect of water molecules on the generated HCHO concentration and background (zero air) concentration was examined. At the outlet of the stripping coil, the sample air was saturated to 100% RH at 10°C. Depending on the inlet air RH (at room temperature) the stripping solution would gain (condensation) or lose (evaporation) water according to the gas/liquid phase equilibrium (Junkermann and Burger, 2006). The bias in calculated HCHO concentrations was within ± 3%.  To check the RH effect on measured HCHO concentration under varying RH conditions, we occasionally compared the observed concentrations with data derived from the certified permeation rate of the permeation tube and the measured gas flow rate (gas phase calibration method). The observed total bias, assumed to characterize the instrument's accuracy

was within ± 5%. Moreover, the instrument's zero air background was observed to increase gradually when increasing RH from 0% to 50%, and decreased when further increasing RH beyond 70%. This could be due to the RH effect on the measurement principle, inlet surface effects, or aging of peristaltic tubes, stripping solution and Hantzsch solution (Kaiser et al., 2014). To account for the influence of changed background on measured HCHO concentration, the background was checked at the beginning and end of each experiment and the measured HCHO concentrations were corrected based on observed background values. On the other hand, well-known ozone interferences (Rodier and Birks, 1994;Kormann et al., 2003) were not of any concern for our studies, as the applied carrier gas was free of ozone."

**Comments and suggestions:**

*The statement "the partial reversibility of HCHO uptake as shown in Fig. 7 can thus be expected as a general feature for various kinds of soils." is a conclusion based on the authors support of the idea that most soils share a similar metal oxide composition. However, I am not entirely sure the claim of various soils showing similar reversibility can be made considering the mechanism the author discusses within this paper do not deal with the chemical composition of the soil, but seem more controlled by surface area effects which could be expected to vary largely between soil types. Surely other factors such as soil pH and surface morphology of a given soil would have a more dominant control on the reversibility of uptake than the metal oxide composition.*

**Responses and Revisions:**

We thank the reviewer's comments. Actually, the uptake mechanism we discussed in the paper was more related to the chemical composition of the soil surfaces, even though some other properties (e.g. surface morphology, soil pH etc.) also contributed. As we found for HCHO uptake on our soil samples, the uptake involved both physical and chemical processes. Each process is assumed to be determined by specific components (reactive sites) on the soil surfaces, which could confer a reasonable explanation why part of the adsorbed HCHO is released back to the gas phase while the other didn't.

Some studies had already proved that HCHO uptake was completely reversible on $SiO_2$, but only partly (< 1~15%) reversible on $\alpha$-$Al_2O_3$ and $\alpha$-$Fe_2O_3$ (Carlos-Cuellar et al., 2003), and the work by Xu et al. (2011) demonstrated oxidation of adsorbed HCHO to dioxymethylene and further to

formate by heterogeneous reactions on the surface of $\gamma$-$Al_2O_3$ particles. We applied the EDX method to analyze the general composition of the soil samples and found that inorganic minerals dominated the soil composition. Among the inorganic minerals, silicon minerals (e.g., silicon oxides) were most abundant followed by aluminum, calcium and iron minerals. Our soil composition and HCHO uptake properties catered to the above reported results and thus we inferred that the uptake of HCHO on our soil surfaces was most probably related to its composition.

Generally, mineral particles occupy the major volume (45% - 49%) of soils (DeGomez et al., 2015). This mineral particle composition, however, can be different depending on the formation process of various types of soils. But still, silicon minerals (e.g., silicon oxides) are the most common mineral particles.

Based on the reviewer's comments, we have made further revision and rephrasing on page 10, line 4:

"Since mineral particles occupy the major volume (approximately 45% - 49%) of soils (DeGomez et al., 2015) and silicon minerals (e.g., silicon oxides) are fairly common in mineral particles, the partial reversibility of HCHO uptake as shown in Fig. 7 may be expected as a general feature for other similar types of soils."

**Comments and suggestions:**

*Page 7, line 19: "Under dry conditions, higher HCHO concentrations", in this case what is the higher HCHO concentrations used?*

**Responses and Revisions:**

We thank the reviewer's comments. We have added the higher HCHO concentration values in page 8, line 13:

"…Under dry conditions, higher HCHO concentrations (20 – 40 ppb) lead to significantly reduced uptake coefficients."

**Comments and suggestions:**

*Page 8, line 15: It seems this sentence is in need of an edit as it reads awkwardly: "…dissociation and desorption equals to that increased in the gas flow per unit time."*

**Responses and Revisions:**

Good suggestion. We have rephrased the sentence on page 9, line 9:

"…the increase in gas flow HCHO concentration equals the HCHO released from the soil due to desorption and/or oligomer dissociation at the same time."

***Comments and suggestions:***

*Page 8, line 16: delete the word 'as' after "coefficient k,".*

**Responses and Revisions:**

Corrected.

***Comments and suggestions:***

*Page 9, line 6: delete the word 'in' after "the soil investigated".*

**Responses and Revisions:**

Corrected.

***Comments and suggestions:***

*Page 10, line 20: delete the word 'the' before "regime III".*

**Responses and Revisions:**

Corrected.

***Comments and suggestions:***

*Page 11, line 26: edit to read "a similar effect".*

**Responses and Revisions:**

Corrected.

***Comments and suggestions:***

*Page 11, line 29: edit "partial" to 'partially'.*

**Responses and Revisions:**

Corrected.

**References:**

Carlos-Cuellar, S., Li, P., Christensen, A. P., Krueger, B. J., Burrichter, C., and Grassian, V. H.: Heterogeneous uptake kinetics of volatile organic compounds on oxide surfaces using a Knudsen cell reactor: Adsorption of acetic acid, formaldehyde, and methanol on alpha-Fe2O3, alpha-Al2O3, and SiO2, J Phys Chem A, 107, 4250-4261, 2003.

DeGomez, T., Kolb, P., and Kleinman, S.: Basic Soil Components: http://articles.extension.org/pages/54401/basic-soil-components, 2015.

Donaldson, M. A., Bish, D. L., and Raff, J. D.: Soil surface acidity plays a determining role in the atmospheric-terrestrial exchange of nitrous acid, Proceedings of the National Academy of Sciences, 111, 18472-18477, 10.1073/pnas.1418545112, 2014.

Junkermann, W., and Burger, J. M.: A new portable instrument for continuous measurement of formaldehyde in ambient air, J Atmos Ocean Tech, 23, 38-45, 2006.

Kaiser, J., Li, X., Tillmann, R., Acir, I., Holland, F., Rohrer, F., Wegener, R., and Keutsch, F. N.: Intercomparison of Hantzsch and fiber-laser-induced-fluorescence formaldehyde measurements, Atmos Meas Tech, 7, 1571-1580, 2014.

Kormann, R., Fischer, H., de Reus, M., Lawrence, M., Bruhl, C., von Kuhlmann, R., Holzinger, R., Williams, J., Lelieveld, J., Warneke, C., de Gouw, J., Heland, J., Ziereis, H., and Schlager, H.: Formaldehyde over the eastern Mediterranean during MINOS: Comparison of airborne in-situ measurements with 3D-model results, Atmos Chem Phys, 3, 851-861, 2003.

Krinke, S.: Experimentelle Bestimmung der Depositionsgeschwindigkeit von Formaldehyd und Ozon über einem Laubwaldbestand, Ph.D, Fakultät Chemie, Universität Stuttgart, 1999.

Rodier, D. R., and Birks, J. W.: Evaluation of Isoprene Oxidation as an Interference in the Cartridge Sampling and Derivatization of Atmospheric Carbonyl-Compounds, Environ Sci Technol, 28, 2211-2215, 1994.

VandenBoer, T. C., Young, C. J., Talukdar, R. K., Markovic, M. Z., Brown, S. S., Roberts, J. M., and Murphy, J. G.: Nocturnal loss and daytime source of nitrous acid through reactive uptake and displacement, Nature Geoscience, 8, 55-60, 10.1038/ngeo2298, 2015.

Xu, B. Y., Shang, J., Zhu, T., and Tang, X. Y.: Heterogeneous reaction of formaldehyde on the surface of gamma-Al2O3 particles, Atmos Environ, 45, 3569-3575, 2011.